# Recent Advances in Chronotherapy Targeting Respiratory Diseases

**DOI:** 10.3390/pharmaceutics13122008

**Published:** 2021-11-25

**Authors:** Keshav Raj Paudel, Saurav Kumar Jha, Venkata Sita Rama Raju Allam, Parteek Prasher, Piyush Kumar Gupta, Rahul Bhattacharjee, Niraj Kumar Jha, Sukriti Vishwas, Sachin K. Singh, Jesus Shrestha, Mohammad Imran, Nisha Panth, Dinesh Kumar Chellappan, Majid Ebrahimi Warkiani, Philip M. Hansbro, Kamal Dua

**Affiliations:** 1Centre for Inflammation, Centenary Institute, Sydney, NSW 2050, Australia; Keshavraj.paudel@uts.edu.au (K.R.P.); n.panth@centenary.org.au (N.P.); 2School of Life Sciences, University of Technology Sydney, Sydney, NSW 2007, Australia; 3Department of Biomedicine, Health and Life Convergence Sciences, BK21 Four, Biomedical and Healthcare Research Institute, Mokpo National University, Muan 58554, Korea; Saurav.balhi@gmail.com; 4Department of Medical Biochemistry and Microbiology, Uppsala University, 75237 Uppsala, Sweden; venkata.allam@imbim.uu.se; 5Department of Chemistry, University of Petroleum & Energy Studies, Dehradun 248007, Uttarakhand, India; parteekchemistry@gmail.com; 6Department of Life Sciences, School of Basic Sciences and Research, Sharda University, Greater Noida 201310, Uttar Pradesh, India; dr.piyushkgupta@gmail.com; 7Kalinga Institute of Industrial Technology (KIIT-DU), KIIT School of Biotechnology, Bhubaneswar 751024, Odisha, India; rahuljonty0798@gmail.com; 8Department of Biotechnology, School of Engineering & Technology (SET), Sharda University, Plot No. 32-34, Knowledge Park III, Greater Noida 201310, Uttar Pradesh, India; nirajkumarjha2011@gmail.com; 9School of Pharmaceutical Sciences, Lovely Professional University, Jalandhar-Delhi Grand Trunk Road, Phagwara 144411, Punjab, India; sukritivns92@gmail.com (S.V.); sachin.16030@lpu.co.in (S.K.S.); 10School of Biomedical Engineering, University of Technology Sydney, Sydney, NSW 2007, Australia; jesus.shrestha@student.uts.edu.au (J.S.); majid.warkiani@uts.edu.au (M.E.W.); 11Department of Pharmaceutics, School of Pharmaceutical Education and Research, Jamia Hamdard, New Delhi 110062, Delhi, India; mohammadimran2024@gmail.com; 12School of Pharmacy, International Medical University, Bukit Jalil, Kuala Lumpur 57000, Malaysia; dinesh_kumar@imu.edu.my; 13Institute for Biomedical Materials and Devices, Faculty of Science, University of Technology Sydney, Sydney, NSW 2007, Australia; 14Discipline of Pharmacy, Graduate School of Health, University of Technology Sydney, Sydney, NSW 2007, Australia

**Keywords:** asthma, chronic obstructive pulmonary disease, pulmonary fibrosis, lung cancer, circadian rhythm, chronotherapy

## Abstract

Respiratory diseases contribute to a significant percentage of mortality and morbidity worldwide. The circadian rhythm is a natural biological process where our bodily functions align with the 24 h oscillation (sleep–wake cycle) process and are controlled by the circadian clock protein/gene. Disruption of the circadian rhythm could alter normal lung function. Chronotherapy is a type of therapy provided at specific time intervals based on an individual’s circadian rhythm. This would allow the drug to show optimum action, and thereby modulate its pharmacokinetics to lessen unwanted or unintended effects. In this review, we deliberated on the recent advances employed in chrono-targeted therapeutics for chronic respiratory diseases.

## 1. Introduction

The lungs are vital organs with a complex architecture, including the tracheobronchial airway tree, airway parenchyma and smooth muscles with a complex microvascular network that efficiently perform gaseous exchange and other essential respiratory functions. Aside from the gaseous exchange process, lungs are constantly exposed to various environmental, household, and occupational irritants that harm the lungs. Recurrent exposure to such irritants an toxins results in the alteration of the lung architecture and abnormal lung function, leading to the development of respiratory diseases [1,2]. Chronic respiratory diseases (CRDs) such as asthma, chronic obstructive pulmonary disease (COPD), cystic fibrosis, and idiopathic pulmonary fibrosis (IPF) are common diseases that affect the airways [3,4,5]. The World Health Organization (WHO) has warned that CRDs are the fourth leading cause of morbidity and mortality in the world and affect approximately one billion people globally, causing an estimated 7.5 million deaths per year, further creating huge global economic, healthcare and social burdens [6]. Current treatment approaches to treat CRDs include anti-inflammatory agents and bronchodilators as the first line of therapy. However, the use of these agents is largely limited due to their potential side effects and unwanted reactions [7,8]. Understanding the complexity and heterogenicity in the pathogenesis of CRDs is essential to nullifying the unmet needs in the treatment of CRDs. The overall survival of organisms and their wellbeing depend on their synchronicity with their internal periodic cycles and the daily rotational cycles of the earth [9]. This synchronicity in mammals is controlled by a time-keeping system, or a biological clock, the suprachiasmatic nuclei (SCN) of the anterior hypothalamus brain region, the master circadian pacemaker of an organism that anticipates and reacts accordingly to the changes occurring in both the external and internal environments [10]. The molecular structure of the SCN consists of both positive and negative feedback loops that are involved in the transcription and translation of different clock-related genes and proteins [11]. The interaction of these loops results in the generation of self-sustained intrinsic physiologic and behavioral rhythms with a recurring periodicity of approximately 24 h, which are called circadian rhythms (CRs). The synchronicity of internal CRs with the daily rotational cycle of the earth is called entrainment. This is mainly controlled by various environmental cues, namely light, social interaction [12], the rhythm of feeding behaviors [13], exercise [14], activity, temperature [15] and the hormone melatonin [16]. These environmental cues are called zeitgebers. Light is the predominant zeitgeber which exerts its influence by the activation of the melanopsin photoreceptor pathway, connecting a subset of retinal ganglion cells to the circadian pacemaker in the SCN. This pathway receives non-visual input called circadian photoreception provided by a subset of intrinsically photosensitive retinal ganglion cells (ipRGCs/mRGCs) that express the photopigment melanopsin and send neural projections to the SCN to reset the phase of the endogenous circadian clock and the linked CRs involved in the physiology and behavior of the organism according to the 24 h cycle [17,18,19].

The CRs are mainly regulated by a core set of various clock genes. These are the basic helix–loop–helix (BHLH) Period–Arnt–Sim (PAS) protein-containing transcription factors such as the ‘circadian locomotor output cycles kaput’ (*CLOCK*) gene and the brain and muscle Arnt-like protein 1 (*BMAL1*) gene [20,21,22]. These genes drive the transcription and translation of various genes and proteins, including three period (PER) genes: PER 1–3, and two cryptochrome (CRY) genes: CRY 1 and 2, to form the transcription-translation feedback loops (TTFLs) that provide the molecular basis of CRs (Figure 1) [23]. In the presence of a zeitgeber, increased transcription of the Bmal1 and clock genes is seen in the nucleus. This increase in the transcription promotes the heterodimerization of BMAL1 protein with CLOCK protein in the cytoplasm. The increased BMAL1-CLOCK heterodimers translocate into the nucleus and bind to Enhancer Box (E-Box) sequences of clock-controlled genes to promote their own recession by increasing the protein synthesis of the negative regulators PER and CRY, which form PER-CRY heterodimers and translocate to nucleus to block the BMAL1-CLOCK-mediated transcription for maintaining the CR [24]. The BMAL-1-CLOCK dimers also regulate other clock-related proteins, including differentiated embryo chondrocyte-1(DEC1) and differentiated embryo chondrocyte-2 (DEC2), a fifth gene clock family which are involved in crosstalk between circadian rhythm and immunity [25]. The BMAL1-CLOCK heterodimer also induces Dec1 via binding to the E-box in the presence of various environmental and inflammatory stimuli, including hypoxia, irradiation and the presence of various cytokines such as tumor necrosis factor-α, which can negatively regulate the expression of DEC2, PER2 and CLOCK-BMAL-1 proteins [26,27]. Similarly, in the presence of various stress factors, including the elevated levels of transforming growth factor-β and polyinosinic-polycytidylic acid, and hypoxia BMAL1-CLOCK dimers also induce Dec2 expression by binding to the E-BOX, which negatively regulates various clock-related gene expression, including DEC1, BMAL1-CLOCK and PER1 [27,28]. During the night cycle, the PER-CRY heterodimer repressor complex degrades via activating the TTFLs to reset the CRs [29]. Other genes, such as REV-ERBs and retinoic acid receptor-related orphan receptor proteins, are also involved in CR oscillation, where they act as counteracting key receptors to maintain CR oscillation by regulating the expression of the BMAL1 gene (Figure 1) [30].

Apart from the central clock that controls the CR output in response to the zeitgebers, numerous peripheral clocks are present in various organs, including the lungs [31], liver [32], spleen [33], kidney [34], heart, fibroblast, muscles, stomach, endocrine glands [35,36], brain and retina [37] with similar molecular architectures as that of the the central clock. The function of these peripheral clocks is to maintain the overall rhythm in the organs by coordinating with the cues emanating from the master pacemaker to regulate the organ function in response to environmental stimuli [38]. Like other organ systems, circadian clock machinery is seen in almost all immune cells, both innate and adaptive, such as monocytes, neutrophils, eosinophils, macrophages, mast cells, dendritic cells and B and T lymphocytes, where light and dark cycles influence their activities [39]. The published literature suggests a bidirectional relationship between the regulation of clock genes and immune cell activation, where innate and adaptive immune mechanisms, including immune cell trafficking, pattern recognition receptor activation in response to external stimuli, phagocytosis and inflammatory mediators’ release are all controlled by the circadian clock [40]. The malfunction of the clock or alterations of CRs lead to disturbed immune responses and rises in the inflammatory status, which further leads to progression of the disease.

Circadian clocks are also seen in the lungs, where they regulate the circadian oscillation of different lung function parameters such as lung resistance, lung functional residual capacity and peak expiratory flow (Figure 2) [41]. CRDs are associated with an abnormal CR of the lungs that reflects the variations in the daily changes of airway caliber, abnormal mucus secretion, increased airway resistance, and decline in lung function, with abnormal immune-inflammatory responses [42]. Disruption to the circadian clock is mainly associated with various environmental irritants and stressors, including allergen exposure, tobacco and biofuel smoke, endotoxin exposure, bacterial and viral pathogens, occupational stresses, which include working night shifts, and travel-associated jet lag [42]. An altered circadian clock leads to circadian-disruption-associated irregular TTFLs, which are further involved in increased levels of oxidative stress, declined lung function and abnormal inflammatory response, commonly seen in the pathogenesis of CRDs (Figure 2) [43]. The primary goal of this review is to study the involvement of circadian clock rhythm in the pathogenesis of different CRDs and therapeutic implications of chronotherapy in the treatment of CRDs.

## 2. Chronotherapy in Asthma

Asthma is an inflammatory airway disease in which the obstruction of airflow occurs due to bronchoconstriction, enhanced mucus secretion leading to inflammation in the bronchial walls. [44]. One of the key features of asthma is that exacerbation gets worse overnight and peaks during early morning. Nocturnal symptoms are usually observed in asthma and are considered important in deciding the timing of the treatment for the management of symptoms. Studies have revealed that nocturnal symptoms such as cough and dyspnea are usually accompanied by circadian alteration in lung function parameters, such as airway inflammation, decline in airflow and airway hyper-responsiveness. The core clock genes that are altered during asthma have recently been characterized, and various therapeutic approaches targeting these genes to control asthma attacks are being investigated [45]. Several drugs, such as leukotriene antagonists, mast cell stabilizers, corticosteroids, anti-IgE antibodies and bronchodilators, have been employed to treat the clinical symptoms of the disease, which are discussed in the subsequent sections [46].

### 2.1. Chronotherapy of Sympathomimetic Drugs

Sympathomimetic drugs, such as salbutamol, terbutaline, bambuterol, salmeterol and formoterol, cause bronchodilation, the inhibition of inflammatory responses and the relaxation of smooth muscles present in the airway. These drugs are available on the market in various dosage forms, such as oral, parenteral and as an inhaler. Inhalers produce a rapid response, whereas oral dosage forms are used for long-term treatment. Parenteral dosage forms are used for the treatment of acute asthma [47]. The pharmacokinetics and pharmacodynamics of sympathomimetic drugs used in the treatment of asthma may vary based on CRs, and this requires extensive investigation to optimize their applications for chronotherapy. Nocturnal asthma, a condition prevalent in two-thirds of asthmatics, demonstrates a variable night-time exacerbation of symptoms, increased airway responsiveness and worsening of lung functions. Symptoms usually manifest between midnight and 8 a.m. and are at their peak at 4 a.m. [48]. Therefore, for the drug to act based on CRs, a delivery system capable of releasing drugs in a pulsatile fashion rather than as a continuous delivery at a predetermined time/site is required. Quereshi et al. developed an oral time-dependent pulsed release system of salbutamol consisting of an effervescent core surrounded by consecutive layers of swelling and rupturable polymers to treat nocturnal asthma. This delivery system aimed to have a lag time of 6 h, i.e., if the medication was taken at bedtime, the system was expected to release the drug after a period of 6 h, i.e., at 4 a.m., when asthma symptoms are more severe [49]. Barnes et al. developed an inhaler of salmeterol with a prolonged duration of action (more than 12 h) that could be more effective in the treatment of nocturnal asthma [50]. Bambuterol is a prodrug of terbutaline that exerts prolonged broncho-dilation for 24 h. A chronotherapeutic trial of bambuterol was conducted to investigate whether the timing (morning versus evening) of administration varied with the duration of action. Interestingly, it was observed that a single daily dose of 20 mg of bambuterol in the evening showed a prolonged duration of action compared with a morning dose [51].

### 2.2. Chronotherapy of Corticosteroids

Glucocorticoids cause anti-inflammatory action and minimize bronchial hyperreactivity. Glucocorticoids produce complete and sustained release action compared with other anti-asthmatic drugs. In one study, Pincus and their coworkers reported that 800 µg of inhaled triamcinolone administrated once daily at 3:00 PM produced a better response as compared to 200 µg of conventional treatment administered four times a day. After a four-week treatment period, the forced expiratory volume in 1 s (FEV_1_) was enhanced up to 77% from 74% upon once-daily dosing of triamcinolone as compared to its conventional form, which was administered four times a day. The results suggested that triamcinolone administered at 3:00 PM did not enhance adverse systemic effects and produced complete relief in nocturnal asthma [52]. Prednisolone is another steroidal drug that is employed to treat asthma. It is a short-acting medication that produces anti-inflammatory effects. In one study, it was revealed that when administrated at night, prednisolone showed a delayed release of up to 22 h, while its administration in the morning showed complete drug release in 4–6 h [53]. In another study, the effect of oral prednisolone was determined in the management of nocturnal asthma, which was compared with a sustained-release theophylline. The findings revealed that overall, the patients’ health improved in the case of a single dose of prednisolone (30 mg) as compared to sustained-release theophylline (600–800 mg). Additionally, various other parameters were evaluated in terms of nocturnal awakenings, salbutamol administration and daytime asthma score. The nocturnal asthma score was drastically decreased in the group of patients that received prednisolone compared to theophylline. In addition, daytime asthma scores were significantly lower in the prednisolone group compared to the theophylline group. Moreover, the FEV 1 was observed to be increased significantly in the group that received prednisolone compared to the group that received theophylline. Overall, this study demonstrated that oral prednisolone was superior compared to theophylline in the management of nocturnal asthma [54].

### 2.3. Chronotherapy of Parasympatholytic Drugs

Drugs such as ipratropium bromide and tiotropium bromide are categorized under parasympatholytic drugs. These are short-acting drugs that reverse airway obstruction and are employed for the treatment of acute and chronic asthma. They are also administered along with sympathomimetic drugs as combination drugs. When they are co-administered with sympathomimetic drugs, their duration of action gets prolonged. Tiotropium bromide is a long-acting bronchodilator, which is used for the treatment of COPD and asthma. It is an antagonist of the M3 acetylcholine receptor, which is present in the respiratory system. Its administration produces smooth muscle relaxation [55]. While drugs are categorized as short acting or long acting, it is important to consider that individual rates of absorption–distribution–metabolism–excretion (ADME) may vary dramatically between patients and depending on the time of delivery. Moreover, there is also a variation in the pharmacokinetics of drugs between pediatric patients and adults, which is due to the difference in gastric pH, gastric emptying rate, intestinal transit, secretion and activity of bile and other fluids, membrane permeability (example: blood–brain barrier) and plasma protein binding liver enzyme [56]. In a double-blind study, a randomized trial conducted with tiotropium bromide with a dose of 18 mg once daily in the morning (9 a.m.) or in the evening (9 p.m.) or an identical placebo concluded that the drug produced sustained bronchodilation throughout the 24 h day without necessarily abolishing the circadian variation in the airway caliber [57].

### 2.4. Chronotherapy of Other Drugs

Theophylline is a phosphodiesterase inhibitor. It shows a weak bronchodilator activity but strong anti-inflammatory action. It enhances the level of CD8+ cells in peripheral blood, minimizes the level of T lymphocytes in the airways and reduces late responses to allergens. Theophylline, at a dose of 5 to 7 mg/kg, is used for the treatment of acute bronchospasm in asthma [58]. The pharmacokinetics of theophylline vary depending on whether the drug is administered in the morning or in the evening. The chronotherapy of theophylline suggests the purposeful delivery of medication in an unequal amount during 24 h, so that the drug can reach their maximum therapeutic concentration during the night-time, when the risk of asthma attack is at its maximum, and a reduced concentration is reached during the daytime, when the risk is minimal [51]. Another drug, a leukotriene antagonist (montelukast), is responsible for restricting the bronchial hyperresponsiveness and airway inflammation by producing anti-inflammatory activities. A double-blind study showed that montelukast improved the FEV_1_ when administered in the evening rather than in the morning [59]. In another study, omalizumab was confirmed to be clinically efficacious in patients with allergic asthma irrespective of age and was reported to significantly improve nocturnal asthmatic symptoms [60]. A summary of drugs used in the treatment of asthma is listed in Table 1.

## 3. Chronotherapy in COPD

COPD is a slowly progressing asymptomatic disease of the lungs which causes irreversible expiratory airway issues. The induction of inflammatory processes in the lungs via cigarette smoking or toxic gases causes decreased elasticity of the alveolar passage, resulting in limited airway space. This narrows the space of the bronchial tree, causing emphysema, and in turn leading to hyperinflation. In addition, hypoxemia develops in the latter stages of COPD, which causes the diaphragm to flatten and ribs to enlarge [68].

The response of patients to a certain formulation in terms of efficacy and acceptability is based on biological timing and endogenous periodicities [69]. The principal action of chronotherapy is based on receptor–ligand interactions that cause the activation of downstream intracellular signaling pathways, the stimulation of effector molecules and increased production of secondary messengers, along with diurnal variations [70]. This therapy also increases the efficiency of modern therapeutic drugs which are dependent on a number of factors, such as controlled drug release or chrono-modulated drug delivery corresponding with the disease rhythm and drug adsorption, targeting the physiological system via chromodynamics and chronotoxicology [71]. The primary therapeutic treatment for COPD includes long-acting inhaled bronchodilators (LAMAs or LABAs). In the case that disease control is not achieved via LAMAs or LABAs, the guidelines then suggest a combination of two treatments [72]. Although there is widespread agreement on the benefits of LAMAs and LABAs in the treatment of COPD, the position of inhaled glucocorticoids in this treatment has been a matter of debate due to their lack of efficacy, safety concerns, and the risk associated with pneumonia [73]. The administration of an inhaled glucocorticoid should be confined to patients with significant loss of lung function and frequent exacerbations based on the guidelines of the Global Initiative for Chronic Obstructive Lung Disease (GOLD) [72].

In 2003, Calverley et al. used tiotropium in the treatment of COPD for the first time in a clinical trial. There was an overall increase in forced expiratory volume 1 (FEV1) throughout the day in both groups in a trial with tiotropium as a single drug either in the morning or evening. Tiotropium caused sustained bronchodilation in the patients without any side effects in the CRs [58]. A similar randomized clinical trial was conducted by Van Noord et al. in 2005, utilizing a combination of tiotropium and formoterol administered daily in patients with COPD. The findings suggested additive benefits on FEV1 over the course of 24 h. The combination of the drugs was highly efficient compared to either of the drugs used as monotherapy and did not cause variability in CRs against COPD [74]. Another clinical trial was conducted in 2006 by Van Noord e. al., using tiotropium and formoterol, where formoterol was administered twice a day and tiotropium once daily. The administration of formoterol in the morning and evening along with tiotropium as a maintenance therapy drastically enhanced the values of forced expiratory volume 1, forced vital capacity and inspiratory capacity in COPD patients [75]. Another clinical trial was conducted by Terzano et al. in 2008, where various combinations of different therapeutic treatments were compared. A randomized clinical trial administered the following combinations to patients with COPD: tiotropium in the morning; tiotropium in the morning and formoterol at night; formoterol twice daily; tiotropium in the morning and formoterol twice daily and tiotropium at night, and formoterol twice daily and tiotropium at night. Combination therapy with tiotropium in the morning and formoterol twice daily was the most beneficial in moderate to severe COPD patients. Treatment with tiotropium in the evening and formoterol twice daily reduced the prevalent night-time symptoms and the use of salbutamol caused less fluctuation in forced expiratory volume 1 for more than 24 h. This showed that the presence of night-time symptoms is dependent on the timing of tiotropium administration [76]. A randomized trial was conducted by Tsai et al. in 2007 to understand the differences in CRs in COPD. During COPD exacerbations, diurnal changes in symptom severity with an increased likelihood of intubation in the early morning hours in COPD patients compared to that of normal patients [77] were observed. A recent study demonstrated the relationship between chronotherapy and certain environmental factors and cigarettes in COPD patients. The circadian molecular clock in COPD patients is either increased or decreased based upon SIRT1, BMAL1 and locomotor activity, which in turn is dependent upon environmental factors or cigarette smoking [78].

Certain commercially available drugs which are used for the treatment of COPD include umeclidinium, a product of GlaxoSmithKline, and glycopyrronium, a product of Novartis Pharmaceuticals UK Ltd. They have become more common in clinical practice recently, and there are no other chronotherapy studies regarding these clinically utilized drugs [69]. So, further studies in future should be mainly focused on these drugs.

## 4. Chronotherapy in Pulmonary Fibrosis

The association of circadian clocks in the pathogenesis of pulmonary fibrosis has been poorly explored. The event of pulmonary fibrosis effectively alters the circadian biology due to the achievement of asynchronous rhythmicity by the alveolar structures owing to the infiltration by fibroblasts. As such, the circadian core clock plays a crucial role in the lung pathophysiology [42] that validates its clinical perspective for managing the disease pathology that eventually forms the basis of chronotherapeutic interventions [79]. Moreover, there is robust evidence obtained from animal models suggesting that the modulation of CRs in the lungs plays a key role in the incidence of developing pulmonary fibrosis in the aftermath of infections caused by parainfluenza and influenza A virus [80].

Vandeleur et al. (2017) reported disruption in CRs in children with cystic fibrosis. The studies were carried out in children belonging to the age group of 7–18 years with cystic fibrosis free from pulmonary exacerbation. The obstructive sleep apnea-18 questionnaire, two-week actigraphy recordings and overnight oximetry formed a major part of the studies. Reportedly, cystic fibrosis contributed significantly towards the lower CRs, in addition to various other factors such as nocturnal cough [81]. Cunningham et al. (2020) reported that the circadian clock protein REV-ERBα inhibits the development of pulmonary fibrosis. Targeting REV-ERBα effectively blocks collagen secretion from human fibrotic lung tissues, thereby making it a desirable therapeutic target in the management of pulmonary fibrosis. In vivo studies on mouse lung fibrosis established fibroblast-mediated strong circadian oscillations. In vitro analysis recognized the association of REV-ERBα with the transcription factor TATA-box binding protein like 1 and the focal adhesions responsible for the activation of myofibroblasts [82]. Vaughan et al. (2019) reported the regulation of the nuclear factor-erythroid 2-related factor (NRF2)/glutathione-directed antioxidant defense pathway via the circadian clock in the modulation of pulmonary fibrosis. Investigations on an in vivo bleomycin treatment-mediated pulmonary fibrosis model discovered a clock-‘gated’ pulmonary response towards oxidative injury, manifesting a severe fibrotic effect on the application of bleomycin at a circadian nadir in NRF2 levels. The lungs of arrhythmic Clock (Δ19) mice indicated lower levels of NRF2 and reduced glutathione, associated with enhanced oxidative damage to proteins, and a spontaneous fibrotic-like pulmonary phenotype. These outcomes established the role of the circadian control of the NRF2/glutathione pathway for countering fibrotic lung damage, thereby promoting novel chronotherapeutic strategies for the treatment of idiopathic pulmonary fibrosis [83].

## 5. Chronotherapy in Lung Cancer

As CR disruption, such as that caused by physiological perturbations (for example, jet lag), shift work, and mutation of the central core clock gene promotes lung cancer progression, chrono-targeted chemotherapy treatment could be a promising therapeutic model for lung cancer treatment [84]. The clock genes are known to influence cell cycle progression by regulating the expression of cell-cycle-related genes such as cyclin D1 (cycD1) and c-Myc; thus, the crosstalk between the core clock gene and cell cycle is involved in cancer progression [85]. c-Myc is a proto-oncogene that plays an important role in cell proliferation and cancer progression; therefore, the inhibition of c-Myc transcription can indirectly promote the upregulation of CycD1 [86]. The mutation in *Per1* and *Per2* results in the deregulation of CycD1 expression, leading to shortening of the cell cycle and an increase in proliferation rate [87]. In recent years, extensive studies have been conducted on CRs in animals and humans, where new promising ideas have been generated for the management of lung cancer chemotherapy. The findings have suggested that apart from “personalized” therapy for molecular pathway inhibition, chrono-targeted anti-cancer treatment should be considered for lung cancer management [88]. Until 2003, lung cancer patients were treated with palliative chemotherapy such as platinum-based combination chemotherapy that improved survival by 8–10 months, while the recent introduction of personalized therapy allows one to choose the best treatment option for each specific lung cancer patient. This is made possible through molecular testing, such as next-generation sequencing that provides information such as main oncogenic drivers (tumor promoter/suppressor) and immune checkpoints that can be targeted with personalized cancer therapy [89,90]. The administration of personalized medications by aligning with a person’s circadian rhythm to improve treatment efficacy and tolerability (reduced side effects) is personalized chronotherapy [91]. The decision to select the optimal timing of personalized chronotherapy and time of the expected target organ/tissue/cells response may vary extensively depending on individual differences in circadian phase (active or resting), that may depend on the intrinsic clock or the individual adjustment to entrainment/zeitgebers [92]. Using machine learning and mathematical modeling approaches, it is now possible to predict optimal times for the delivery of chrono-therapeutics. Personalized cancer treatments based on the biological diurnal rhythm of the patient show decreased side effects and improved treatment success [93]

The aim of chronotherapy is to deliver a therapeutic regimen when the biological clock of the normal/healthy cell is at rest so that these cells are protected from treatment-related unwanted effects or toxicity. As cancer cells undergo uncontrolled growth (cell division), they are more vulnerable to the chemotherapeutic regimen [94]. Thus, chronotherapy primarily works by identifying the rest phase of healthy cells and CRs of tumor cells [88]. The dose response of the anti-cancer therapy is dependent on the circadian clock to modulate the cell cycle, DNA repair and apoptosis. However, to date, the in-depth mechanisms that reveal the link between circadian biology and the molecular action of anti-cancer treatments are not fully understood. This has led to limitations in the potential use of CR-based therapy (chronotherapy) in clinical settings.

The identification of cancer invasion/metastasis-related core circadian rhythm genes and proteins is crucial for the therapeutic management of lung cancer. Chen et al., 2020 reported that circadian gene hepatic leukemia factor (HLF) expression was significantly decreased in early relapsed NSCLC tissues, and this decreased expression was correlated with early progression as well as metastasis in NSCLC patients. This was further supported by the study of lung colonization and metastasis of the bone, liver and brain where upregulating HLF inhibited and silencing HLF promoted all the processes of colonization and NSCLC metastasis. Furthermore, the downregulation of HLF expression promoted the proliferation of NSCLC cells by activating NF-κB/p65 signaling through interfering with the translocation of PPARα and PPARγ. Further studies found that both methylation and genetic deletion promote the downregulation of HLF in NSCLC tissues. This suggests that the circadian gene HLF may serve as a novel target to inhibit metastasis and may be useful in the management of NSCLC [95]. A recent study by Bellet et al., 2021 demonstrated that one of the core circadian clock proteins, period 1 (PER1), and the tumor suppressor protein p53 negatively cross-regulate each other’s activity and expression to modulate the sensitivity of tumor cells to chemotherapy. PER1 interacts with p53 to decrease its stability and impede its transcriptional activity, while p53 represses the transcription of PER1. This study showed that PER1 decreased the sensitivity of tumor cells to chemotherapy-induced apoptosis, both in vitro and in xenograft mice model (in vivo), suggesting the detailed understanding of the association between the circadian clock protein/gene and tumor regulatory proteins/gene. This has also become the rationale for future research regarding progress in chronotherapy [96]. The airways in the respiratory tract express various transporter genes, such as the ABC family. More specifically, abcc2 protein or *abcc2* gene expression show rhythmicity over the 24 h circadian oscillation, where gene expression ranges from 3-to-6-fold and protein expression from 2-to-3-fold [97]. The metabolism of various chemotherapeutic drugs, such as cisplatin, epirubicin, docetaxel, doxorubicin, paclitaxel, vinblastine and methotrexate, are associated with ABC family transporters because abcc2 transporters are responsible for effluxes of these chemotherapeutic agents. As these transporters are present locally in the pulmonary system, there is a clear role of influence of the circadian clock gene on the efficacy of these chemotherapeutic agents. In this context, the time of administration of chemotherapeutics to patients also plays a key role in determining the efficacy of the treatment. For example, it was found that the efficiency of 5-flouoracil was augmented when it was administered early in the morning at 4 a.m. Similar results were found with regard to irinotecan at 5 am and oxaliplatin at 4 p.m. [98,99]. The efficiency of 5-fluoracil (5-FU) was related with increased intracellular dehydropyrimide dehydrogenase (DPD) enzyme activity in healthy cells as well as low DNA synthesis in healthy tissues. In the case of oxaliplatin, its efficiency was related with significantly decreased glutathione (GSH) levels. Likewise, the efficiency of irinotecan was related with increased Bcl-2 gene expression and decreased S phase cells in healthy cells. DPD is an enzyme involved in the metabolism of 5-FU and GSH. DPD is found in the cytoplasm of most cells as tetrapeptide. It was revealed that the levels of DPD were at their peak between midnight and 4:00 a.m. and the levels of GSH peaked near noon. Comparatively, the level of GSH was found to be higher in bone marrow compared to other organs and its levels was associated with the reduced toxicity of the anti-cancer drug “oxaliplatin” [100].

Various promising compounds have been studied for their potency against lung cancer through the modulation of circadian clock gene/proteins. Epigallocatechin-3-gallate (EGCG) is one of those potential compounds studied for its ability to inhibit self-renewing lung cancer stem-like cells through the inhibition of CLOCK in NSCLC cell line A549 and H1299. The expression of the CLOCK gene and protein levels were significantly increased in A549 and H1299 spheroids compared to their parental cells. Interestingly, the knockdown of CLOCK by siRNA resulted in a reduced ability for sphere formation and the inactivation of the Wnt/β-catenin signaling pathway, while treatment with EGCG repressed CLOCK expression in the spheroids. Moreover, in a xenograft model, EGCG suppressed the cancer stem-cell-like features (CD133, CD44, Sox2, Nanog and Oct4 protein) of A549 and H1299 cells by targeting CLOCK, suggesting EGCG is a compound with chronotherapeutic potential to inhibit the self-renewal capacity of lung cancer stem-like cells by modulating CLOCK [101]. In another study, it was observed that doxorubicin (DOX) altered the core clock genes and various cytokines in macrophages extracted from tumor (Lewis lung carcinoma)-bearing mice. In the tumor tissue, the authors observed a disruption of circadian rhythm, as shown by a decrease in F4/80 (a macrophage surface marker) at the end of the dark period (ZT 22: 4 a.m.) and increased CD11c (transmembrane protein found in macrophage) expression in the middle of the light period (ZT 6: 12 p.m.). The analysis of clock genes at six different time points (ZT02: 8 a.m., ZT06: 12 p.m., ZT10: 4 p.m., ZT14: 8 p.m., ZT18: 12 a.m. and ZT22: 4 a.m.) in peritoneal macrophage stimulated with lipopolysaccharide (LPS) and treated with DOX for 24 h showed that DOX decreased Clock and Per1 expression. Likewise, DOX also decreased the tumor necrosis factor (TNF)-α protein expression after 6 and 24 h, while interleukin (IL)-1β protein expression only decreased after 24 h. This suggests that DOX treatment could be beneficial in inhibiting LPS-induced inflammatory cytokines and in slowing down inflammation associated with cancer progression [102].

## 6. Clinical Studies on Chronotherapy in Various Respiratory Diseases

Various clinical trials have been carried out to evaluate the potential of drugs in the management of respiratory diseases such as asthma, lung cancer, COPD and cystic fibrosis (Table 2). A randomized controlled trial was conducted to characterize the circadian rhythm of lung function using a database of 3379 FEV1 values from 189 patients with mild or moderate asthma receiving ICS. This study predicted that the lowest FEV1 value would be observed in the early morning and the highest FEV1 value would be observed in the early afternoon, with a population mean fluctuation of 170 mL. This finding was consistent with another study that reported that the symptom of asthma exacerbation usually occurs early in the morning [103]. Another randomized control trial conducted on 20 mild–moderate asthma patients studied whether a once-daily inhaled formoterol in the evening, administered from the combination budesonide/formoterol (BUD/F) Turbuhaler, notably ameliorated the circadian rhythm in airway tone for over 24 h. Patients received either inhaled BUD/F (2 × 100/6 microg) or a placebo at 8 p.m. on two separate occasions followed by measurement of lung function parameters, such as FEV1, airways conductance and maximum expiratory flow at baseline, at 1 h after administration and every 4 h post-administration for the next 24 h. The results showed that BUD/F remarkably ameliorated all three lung function parameters throughout the 24 h period, with a difference in FEV1 at 24 h of 0.20 L (0.04–0.35 L) as compared to the placebo control. BUD/F also ameliorated the biphasic pattern of the circadian rhythm in the airway, suggesting that a once-daily evening dose of formoterol showed prolonged bronchodilation for at least 24 h [104]. A randomized control trail compared the pharmacokinetics and toxicity of tobramycin administered in the morning versus evening for exacerbations of cystic fibrosis in children (n = 18) aged 5 to 18 years. The morning dose was administered at 8 a.m., and the evening dose was administered at 8 p.m. As aminoglycosides such as tobramycin are known to cause nephrotoxicity, the primary focus of this study was to evaluate whether the timing of administration differentially affects/impairs the renal function by measuring the serum levels of tobramycin after 1 h and 3.5–5 h of infusion. Melatonin levels were measured in saliva samples at different time points (6 p.m., 7 p.m., 8:30 p.m., 9:30 p.m., 12 a.m., 6 a.m. and 12 p.m.) and changes in the circadian rhythm were evaluated by measuring melatonin via a radioimmune assay. Although there were no remarkable differences in renal clearance between the two groups, the increase in urinary KIM-1 (a marker of renal toxicity) was observed to be higher in the evening administration group to that of morning administration group, suggesting evening-administered individuals possess a greater risk of nephrotoxicity. Normal circadian rhythms (referring to dim light melatonin onset between 6 and 10PM and/or a melatonin peak during the night) were observed in 7/11 participants (64%), while the remaining four showed disruptions in their circadian rhythms and rises in melatonin levels [105]. Another clinical trial explored the circadian variability in terms of pharmacokinetics of total and unbound cisplatin administered in the morning at 6 am and in the evening at 6 pm in NSCLC patients. The results showed that the clearance of total and unbound cisplatin was primarily dependent on body surface area. As compared to 6 am cisplatin administration, the clearance of total and unbound cisplatin was increased by 1.38 and 1.22-fold, respectively, when administered at 6 p.m., suggesting that circadian rhythm can possibly influence the metabolism of cisplatin [106] Another study by this group (Li et al., 2015) compared the superiority (in terms of relieving side effects) of cisplatin-based chronotherapy (in 17 NSCLC patients) versus a routine chemotherapy group (in 24 NSCLC patients). Although there was no remarkable difference in the total response rates between the groups, the incidence of leucopenia and neutropenia in the chronotherapy group was more notably reduced than that in routine chemotherapy. Similar, occurrences of gastrointestinal discomfort such as nausea in the chronotherapy group were remarkably lower compared to the routine chemotherapy group, suggesting the advantage of cisplatin-based chronotherapy in relieving the noxious effects of chemotherapy [107].

## 7. Lung-on-a-Chip Circadian Research Platform

Microfluidics is the science and technology that can be utilized to mimic different tissues or organs by precisely patterning and co-culturing different cell types in a well-controlled microenvironment [109]. Microfluidic systems offer the automation of assays, biocompatibility and real-time microscopy with high resolution. Furthermore, microfluidics can be integrated with the spatial and temporal control of the physical environment (surface, matrix, mechanical strain and flow-induced shear stress), chemical environment (drugs, nutrients, hormones, enzymes and toxins) and electrodes for localized sensing and controlling. Moreover, dynamic cellular processes (circadian rhythm, signaling dynamics, metabolism and pharmacokinetics) can also be monitored [110].

Recently, organ-on-chips have also been tested to incorporate the factors that mimic circadian rhythms in organs, which is vital when the physiological impacts of different drugs are being tested [111]. Organ-on-chip models can be fitted with pumps and valves to maintain a time-dependent concentration of drugs or hormones to be perfused in the targeted channel [112]. Time-division multiplexing can also be incorporated to support the delivery of multiple doses of drugs at the same or different time intervals. A daily rhythm cycle can be modeled to mimic significant parameters of drug metabolism in end-organ tissues and provide valuable information for developing new and effective therapeutics. By incorporating rhythmic endocrine regulation, these systems can be used to model and study the advanced effects of drug–organ–hormone interactions in humans [111]. Different organ-on-chip models representing different organs of the human body can also be interconnected to form multisystem homunculi, which could provide an insight into the dynamics of complex physiological interactions between different organs and circadian rhythms not accessible due to practical and ethical concerns in clinical trials [111]. The possibility of developing personalized chips using samples derived from individuals can further aid in recapitulating the physiology of a specific individual [113]. Given the numerous advantages and incorporation of the circadian rhythm, lung-on-a-chip models can enhance the development of effective therapeutics with the adjustment of doses, methods, and timings of the administration of drugs to treat respiratory diseases with maximum efficacy.

## 8. Chronotherapy in COVID-19

The COVID-19 pandemic means that the discovery of safe and effective therapies for SARS-CoV-2 is urgently needed. The understanding of chronobiology with regard to how COVID-19 infection affects circadian rhythms may provide opportunities for the development of chronotherapy for SARS-CoV-2. The host immune responses in COVID-19, particularly the severe inflammation that can possibly cause multi-organ failure, strongly modulate disease severity. The activity of all components of the immune system, including the inflammatory immune responses, follow robust CRs [114]. It is predicted that the severity of COVID-19 infection depends on the time (day or night) when infection occurs, because the harm caused by a rapidly replicating virus and the way it is offset by our immune system depends on the phase of the circadian rhythm of the host. This can be confirmed by both in vitro and in vivo experimental setups where human cell lines (in vitro) and mouse models (in vivo) are infected with SARS-CoV-2 in both the active phase and resting phase (12 h apart) [115].

In COVID-19 cases, a group of patients experience cytokines release storm (CRS) that results in acute respiratory distress syndrome (ARDS). Since the intrinsic circadian clock of the lungs, together with the immune system, regulate the various component of CRS, the use of chrono-targeted therapy can be very effective in the management of ADRS during COVID-19 infection [86]. As there is variation in the circadian rhythm in lungs and the immune system (between healthy and diseased), the effect of immune metabolic modulators or anti-inflammatory drugs on cells or cytokines release from diseased cells/tissues also depends on the timing of administration [116]. It is likely that the appropriate selection of anti-inflammatory chronotherapy can be beneficial to combat the detrimental CRS and ADRS during COVID-19 infections. For example, chronotherapy can differentially alter the level of different cytokines such as chemokine (C-X-C motif) ligands (CXCL10), IL-1β, IL-4, IL-8, IL-10, TNFα and Toll-like receptors during viral infection, including COVID-19 [80,116]. Following this observation, a recent investigation on a murine bone injury (tibia fracture) model showed that anti-inflammatory cytokines such as IL-13 and IL-4 and clock genes such as *Per2* could be effectively regulated by administrating non-steroidal anti-inflammatory drugs during the active phase of the circadian rhythm [116]. Interestingly, the cytokines that were modulated by the timing of drug administration coincide with cytokines such as IL-1β, IL-8, IL-10R, IL-6R and TNFα that are involved in the CRS in COVID 19.

Circadian clocks regulate the pharmacokinetics, which includes absorption–distribution–metabolism–excretion and the potency of many medications, because various drug targets and genes or proteins that act as drug transporters or facilitate biotransformation show daily rhythmic oscillation [117]. Investigation of the optimized dosing time regarding dynamic host–pathogen interactions could significantly enhance the therapeutic activity of medicines, including vaccines. Therefore, researchers or pharmaceutical industries must consider circadian rhythms when designing and dosing drugs of choice and vaccines for SARS-CoV-2 to achieve the optimum clinical outcomes. A recent study revealed various host factors and physiological pathways that could be promising drug targets for COVID-19 [115]. This study used affinity purification mass spectrometry to identify 332 different human proteins that interact with SARS-CoV-2 proteins and found the mouse orthologues of these 332 SARS-CoV-2-interacting host factors using mouse circadian transcriptome data. Interestingly, it was revealed that the expression of 30% of these host factors shows circadian oscillation. This study provides the basic understanding of the crosstalk between circadian clocks and viral infections and the fact that we are more vulnerable to certain respiratory viruses during the early morning [118]. A preliminary retrospective study was conducted in Ferrara, Italy to compare (morning vs. evening administration) the possible effect of an antiviral (darunavir–ritonavir) single daily dose for 7 days to COVID-19 patients. Interestingly, it was found that morning therapy was able to remarkably reduce the C-reactive protein (a marker of inflammation) [119]. A similar observation was found in flu vaccines as well, where it was found that antibody response to flu vaccines is stronger when vaccinated in the morning than in the afternoon [120]. Zhang et al. 2021 investigated the immunological response to the SARS-CoV-2 vaccine developed by Sinopharm, Beijing delivered to 63 healthcare workers either in the morning (9 a.m.–11 a.m.) or afternoon (15 p.m.–17 p.m.) on day 0 and day 28. Interestingly, the data revealed that vaccination in the morning leads to a stronger immune response to the vaccine compared to the afternoon [121]. Through these studies, we can speculate that there is a crucial role of chronotherapy in the safe and effective management of COVID-related immunological response, inflpmmation and other symptoms.

## 9. Future Perspectives

Despite the considerable advancement in the field of circadian rhythms and chronotherapy, significant information-related gaps still exist between patients receiving treatment, community pharmacists and physicians based on prescription-related specifics, awareness of circadian rhythms, application, and attitudes toward the application of chronotherapy in practice [122,123]. Therefore, physicians and community pharmacists could both play a key role in providing appropriate information to patients on optimal dosing times for effective clinical implementation and achieve maximum benefits of this therapy. Notably, through the application of novel drug delivery technologies, i.e., modulating the release of drugs in a pulsatile fashion (delivering an immediate pulse of drug after a delay, so that the maximum of the drug is released at the suitable timing) or maintaining the necessary drug concentration in plasma for 24 h through a single dose is highly desirable for chronotherapy. Additionally, preliminary analysis and screening of newer drugs during preclinical studies for their chronotherapeutic potential could be an effective method to enhance research and development in pharmaceutical industries [122]. Furthermore, elucidating molecular pathways that precede with time-based worsening of symptoms and defining the role of the molecular clock in these pathways could provide a novel therapeutic target in future. In addition, chronopharmacology (chronopharmacokinetics and chronopharmacodynamics) are other essential parameters that must be well-monitored and studied for adequate clinical responses. Additionally, randomized trials suggested that the study design for this module of treatment missed the optimal timing elucidation, excessive or insufficient dose, or interpatient differences masked the circadian timing effects. Therefore, in future, chronotherapy requires a new methodology for the design of clinical trials, in which each individual patient would receive individualized chronomodulated therapies computed by data-driven mathematical models for the appropriate clinical validation of the treatment. Another challenge existing with this therapy concerns how to consider the biological rhythms of each patient [124]. An accurate characterization of the internal circadian time is the prerequisite for this therapy. Therefore, advancements in technology for measuring biological time, the general assessments of circadian rhythmicity, and the correct use of this rhythmic variable to predict biological times are the most essential developments in future and will decide the fate of this therapy. The application of machine learning algorithms in combination with predictive mathematical models of regulatory networks in circadian research could enable us to utilize and extrapolate biological data and to identify predictive circadian parameters more accurately [93]. This knowledge may lead to the identification of biomarkers within circadian-expressed genes and can ultimately be used for the accurate estimation of drug administration timing in diseases [93].

## 10. Conclusions

In-depth understanding of how the disruption of circadian rhythm leads to the progression of respiratory diseases is essential for the selection of a chronotherapy for a specific individual. Moreover, defining the influence of the core circadian clock in the cell signaling pathways leading to lung disorders could unlock novel therapeutic avenues. The efficacy and side effects of a therapy may vary when administration occurs at 8 a.m. compared to 8 p.m.; therefore, elucidating the circadian oscillation (time of peak and trough) of various disease-related proteins, genes and enzymes will facilitate the selection of the timing of drug administration. For various diseases including respiratory diseases, chronotherapy could be superior over routine therapy in terms of efficacy and the control of side effects. However, more scientific studies that support chronotherapy as a choice of therapeutic management over conventional therapy are essential for the validation of its effectiveness.

## Figures and Tables

**Figure 1 pharmaceutics-13-02008-f001:**
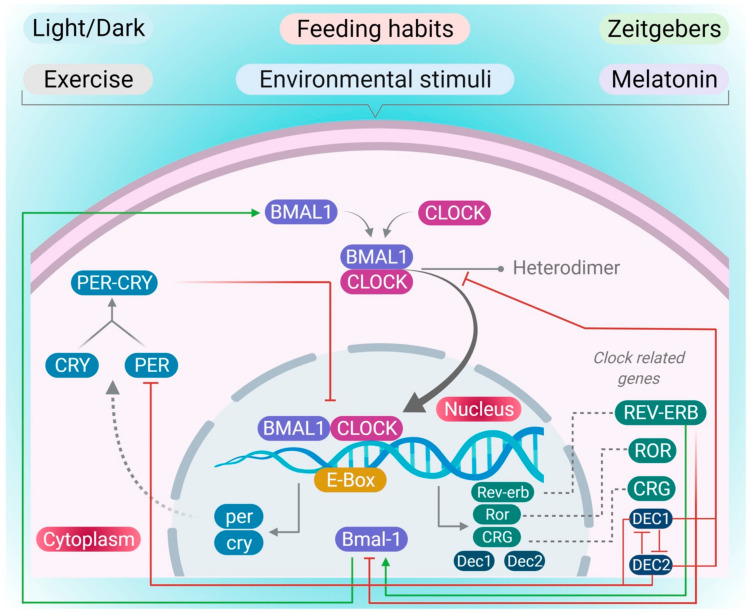
Circadian clock rhythm machinery: In the presence of various environmental, behavioral and environmental stimuli, BMAL1 and CLOCK will interact to form the heterodimer that translocates into the nucleus to induce the transcription of various genes, including per, cry, Rev-erb, ror, Dec1 and Dec2 and clock-related genes which regulate the CR.

**Figure 2 pharmaceutics-13-02008-f002:**
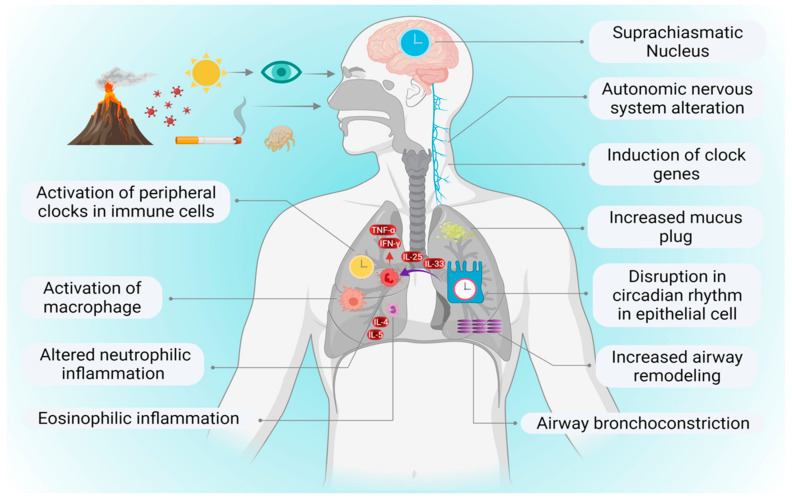
Association between central and peripheral CRs clocks to activate immune system in chronic respiratory diseases: Light/dark cycle in the presence of various environmental irritants disrupts both the central circadian clock located in the suprachiasmatic nucleus in the brain and peripheral circadian clock located in the lungs and other immune cells. The association between both the central and peripheral clocks leads to activation of various immune cells to regulate various symptoms involved in the pathogenesis of different chronic respiratory diseases.

**Table 1 pharmaceutics-13-02008-t001:** Drugs used in the treatment of asthma.

Drug	Findings	Reference
Salbutamol	Significant improvement was observed in FEV1 levels in the morning in asthmatic patients; however, no such improvement was recorded in COPD patients. No significant impact of salbutamol CR was observed in the alteration of sleep patterns and oxygen levels in asthmatic and COPD patients.	[61]
Terbutaline	No alteration in sleep pattern was revealed after the administration of oral terbutaline. It improved the morning peak flow and decreased inhaler usage at night. Lower numbers of awakenings at night were observed with oral terbutaline.	[62]
Bambuterol	It significantly decreased the nocturnal asthmatic symptoms almost by 60-fold. It was preferred by 49% of nocturnal asthmatic patients compared to 36% who preferred salbutamol. It exhibited less severe side effects as compared to salbutamol.	[63]
Salmeterol	It improved the patients’ life quality, decreased night awakenings and reduced nocturnal arousals. Notably, daytime cognition was not altered through the treatment of salmeterol As compared to theophylline, more significant outcomes were observed in the case of salmeterol.	[64]
Ipratropium bromide	It decreased the morning “dip” of the peak in the flow rate of expiration. It significantly improved nocturnal asthma symptoms. Ipratropium bromide alone demonstrated significant reductions in the severity of asthma as compared to the combination of salbutamol and ipratropium.	[65]
Tiotropium bromide	Morning and evening peak expiratory flows were significantly improved in the case of tiotropium bromide as compared to placebo groups. FEV_1_ was significantly improved at all time points during the whole observation period in the case of tiotropium bromide compared to placebo. No adverse events were reported with regard to tiotropium bromide as compared to placebo.	[58]
Hydrocortisone	FEV1 was observed at two time points, i.e., 04:00 h and 16:00 h. FEV1 values were reported to be higher at all points of time in children with nocturnal asthma.	[66]
Prednisolone	It exhibited a significant improvement in nocturnal asthma symptoms. The mean nocturnal awakening was reduced by almost 83% in the case of delayed-release prednisone as compared to conventional ones.	[53]
Omalizumab	The rate of severe exacerbations was significantly reduced by almost 69% and 75% in subjects aged > 50 and aged < 50, respectively. It significantly improved the nocturnal asthmatic symptoms and awakenings. It was established as an effective therapy in the case of severe allergic asthma.	[60]
Sodium cromoglycate	Improvement in nocturnal oxygenation was observed. It was revealed that the degranulation of mast cells might not be important regarding the cause of nocturnal asthma. Downfalls in FEV_1_ and FVC were reported in all the patients at night. The treatment of sodium cromoglycate observed no alteration in breathing pattern.	[67]

**Table 2 pharmaceutics-13-02008-t002:** Clinical study of drugs modulating circadian rhythm in respiratory diseases.

Drug	Findings	Disease	Reference
Inhaled corticosteroids (ICS)	In asthma patients receiving ICS, the lowest FEV1 value was observed in the early morning and the highest FEV1 value was observed in the early afternoon, with a population mean fluctuation of 170 mL.	Asthma	[103]
Budesonide/formoterol combination inhaler	In comparison to the placebo control group, BUD/F remarkably ameliorated the lung function parameter throughout the 24 h period. BUD/F also ameliorated the biphasic pattern of the circadian rhythm in the airway, suggesting a once-daily evening dose of formoterol showed prolonged bronchodilation for at least 24 h.	Asthma	[104]
Tobramycin	No remarkable changes were observed in renal clearance between morning and evening administration of tobramycin. The increase in urinary KIM-1 (a marker of renal toxicity) was higher in the evening administration group compared to the morning group. There was normal circadian rhythm in 7/11 participants, while the remaining 4 showed disruptions in their circadian rhythms and rises in melatonin levels.	Cystic fibrosis	[105]
Cisplatin	Circadian rhythm could influence cisplatin metabolism, suggesting the conventional dose adjustment of cisplatin based on body surface area.	Lung cancer	[106]
Cisplatin	Cisplatin-based chronotherapy was beneficial, with fewer side effects compared to routine chemotherapy.	Lung cancer	[107]
Tiotropium bromide	Tiotropium improved the mean FEV1 (over 24 h) and nocturnal FEV1 (at 6-week visit) in the morning and evening groups compared with the placebo. Tiotropium resulted in prolonged bronchodilation for 24 h without necessarily impairing circadian variation in airway caliber.	COPD	[58]
Tiotropium with/without formoterol	At baseline, there was circadian variation in FEV(1), FVC, and IC, and this was maintained throughout the treatment periods. Tiotropium alone improved the average FEV(1) by 0.08 L, while the combination of tiotropium and formoterol (a once-daily dose) improved FEV(1) by 0.16 L and the combination with a twice-daily dose (morning and evening) of formoterol improved FEV(1) by 0.20 L. In comparison to tiotropium alone, combination therapy in the morning resulted in the improvement of FEV(1), FVC and IC for more than 12 h. The combination therapy with a twice-daily dose of formoterol further improved FEV(1) for 12 h, while FVC and IC were improved for less than 12 h. In comparison to tiotropium alone, combination therapy (with both once- and twice-daily doses) resulted in a decrease in the need for rescue salbutamol during the daytime. The twice-daily dose of formoterol reduced the need for salbutamol during the night-time as well.	COPD	[108]

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
