# Peer review of "Recent Advances in Chronotherapy Targeting Respiratory Diseases"

_pharmaceutics, 2021, doi:10.3390/pharmaceutics13122008_

Round 1

Reviewer 1 Report

    A paper is a review of the “Recent advances in chronotherapy targeting respiratory diseases” – both aspects, chronotherapy, and respiratory diseases are highly topical to today’s agenda. I found this paper comprehensive, attractive, and reader-friendly while reviewing and believe it should be interesting to a wide auditory of readers.
    The issues provided below are intended to help authors to improve their manuscripts before publishing.
    P.2. L. 77-79: Authors introduce circadian entrainment (“synchronicity of the internal circadian rhythms with the daily rotational cycle”) with the following sentence: “This is mainly controlled by the various environmental cues namely, light, aging, social interaction [10], feeding activities, availability of food [11], exercise [12], activity, temperature [13] and the hormone melatonin [14].”
    This sentence should be corrected since 1. Aging does not belong to environmental cues for the circadian time system, 2. The term “feeding activities” does not suit well as it stands as availability of food is mentioned next, and activity is mentioned later.
    P.2. L. 80-84: “Light is the predominant zeitgeber which exerts its influence by the activation of melanopsin photoreceptor pathway. This pathway receives the non-visual input called circadian photoreception from the rod-cone system and sends neural projections to the SCN to reset the phase of the endogenous circadian clock and the linked CRs involved in the physiology and behavior of the organism according to the 24-hour cycle.”  This sentence must be corrected: non-visual input that provides circadian photoreception is provided by a subset of intrinsically photoreceptive retinal ganglion cells (ipRGs/ mRGCs) that express the photopigment melanopsin and despite receiving input from the classical photoreceptors, the cones, and rods (DOI: 10.1016/j.preteyeres.2012.03.003 ;  10.1016/j.cub.2019.11.031), are not equivalent to rods and cones. Authors must specify the role of intrinsically photoreceptive (melanopic) ganglion cells that are distinct from rods and cones as far as circadian entrainment is considered.
    P.2. L. 113-115: Authors refer to “apart from the central clock that controls the CR output in response to the zeitgebers, numerous peripheral clocks are present in various organs including the lungs [21], liver [22] and the spleen [23] with similar molecular architecture as seen in the central clock”. Why are examples limited exactly to these organs? At least, kidney (doi: 10.1681/ASN.2010080803), heart, fibroblasts, muscles, stomach, endocrine glands (https://doi.org/10.1016/j.neuron.2012.04.006; doi: 10.1146/annurev-neuro-060909-153128 ) brain and retina (doi: 10.1002/bies.20777) should not be omitted in this agenda.
    P.2. L. 113-115: “CRDs are associated with abnormal CR of the lungs that reflect the variations in the daily changes of airway caliber, abnormal mucus secretion, increased airway resistance, insomnia, and decline in the lung function with abnormal immune-inflammatory responses [28].” I feel like mentioning insomnia logically does not belong here.
    P.5. L. 175-176 – Please correct text formatting.
    P.9. L. 338 – Please explain the term OSA-18 as it is mentioned once, but never defined.
    P.9. L. 347 - Please explain the term TBPL-1 as it is mentioned once, but never defined.
    P.9. L. 347 - Please explain the term NRF-2 as it is mentioned once, but never defined.
    P.10. L. 362 – “As CRs disruption promotes lung cancer progression, chrono-targeted chemotherapy treatment could be a promising therapeutic model for lung cancer treatment [75]”. Please define exactly, what kind of “CRs disruption” is meant here, since CRs disruption may have different causes: (artificial) light by night, shift-work, age-associated, clock gene mutations, etc.
    The term “personalized” chronotherapy was mentioned thrice with refs 76,96,98, though it never was substantiated. Authors are advised to define what they mean by “personalized chronotherapy” or refer to original concepts in the papers cited.

    Overall:
    While discussing optimal timing of delivery and timing of the expected target organ/tissue/cells response, please consider that it may vary extensively depending on broad in time individual differences in circadian phase, that may depend on the intrinsic clock or the individual adjustment to zeitgebers, the basics of personalized chronotherapy.
    Also, while discussing “short-term”, and “long-term” drug formulas, please note that individual rates of absorption – distribution – metabolism – excretion (ADME) may vary dramatically between patients and depending on the time of delivery.
    I understand how tempting is the concept of standardizing timing for the given chemical – once and for all. However, personalized timing is highly important, since discarding adjustment of personalized phase differences in chronotherapy may negate its basic idea.

    Since the paper’s title is “Recent advances in chronotherapy targeting respiratory diseases” paying at least some attention to chronotherapy of COVID-19 associated pulmonary disease, or drugs that may be repurposed to prevent COVID-19 caused pulmonary complications or combat its risks were expected to be found in this review.

Author Response

Author response to reviewer comments

The authors are very thankful to the Editor, the Editorial team, and the Reviewers for consideration of our manuscript and for providing their valuable suggestions.

We have now addressed all the comments and incorporated the required changes into a revised version (highlighted in red font) as described in the point-by-point response below. These changes have helped to improve the manuscript substantially, which, we now hope is acceptable for publication in the Pharmaceutics journal.

Reviewer: 1

Comments to the Author

A paper is a review of the “Recent advances in chronotherapy targeting respiratory diseases” – both aspects, chronotherapy, and respiratory diseases are highly topical to today’s agenda. I found this paper comprehensive, attractive, and reader-friendly while reviewing and believe it should be interesting to a wide auditory of readers. The issues provided below are intended to help authors to improve their manuscripts before publishing.

Author response: We are extremely grateful for reviewer 1 constructive comments and going in detail throughout the manuscript. Indeed, the critical comments/suggestions were very helpful to improve the quality of  our manuscript.

    P.2. L. 77-79: Authors introduce circadian entrainment (“synchronicity of the internal circadian rhythms with the daily rotational cycle”) with the following sentence: “This is mainly controlled by the various environmental cues namely, light, aging, social interaction [10], feeding activities, availability of food [11], exercise [12], activity, temperature [13] and the hormone melatonin [14].”

  This sentence should be corrected since 1. Aging does not belong to environmental cues for the circadian time system, 2. The term “feeding activities” does not suit well as it stands as availability of food is mentioned next, and activity is mentioned later.

Author response: We agree with reviewer comments. We have now removed the word aging and we have changed feeding activities, availability of food to rhythm of feeding behaviors to align with cited reference.

Location: page 2, line 78

    P.2. L. 80-84: “Light is the predominant zeitgeber which exerts its influence by the activation of melanopsin photoreceptor pathway. This pathway receives the non-visual input called circadian photoreception from the rod-cone system and sends neural projections to the SCN to reset the phase of the endogenous circadian clock and the linked CRs involved in the physiology and behavior of the organism according to the 24-hour cycle.”  This sentence must be corrected: non-visual input that provides circadian photoreception is provided by a subset of intrinsically photoreceptive retinal ganglion cells (ipRGs/ mRGCs) that express the photopigment melanopsin and despite receiving input from the classical photoreceptors, the cones, and rods (DOI: 10.1016/j.preteyeres.2012.03.003 ;  10.1016/j.cub.2019.11.031), are not equivalent to rods and cones. Authors must specify the role of intrinsically photoreceptive (melanopic) ganglion cells that are distinct from rods and cones as far as circadian entrainment is considered.

Author response: We agree with the reviewer comments and appreciated the suggestion with reference paper. We have now revised the sentence and cited the suggested references.

Location: Page 2, line 80-84

    P.2. L. 113-115: Authors refer to “apart from the central clock that controls the CR output in response to the zeitgebers, numerous peripheral clocks are present in various organs including the lungs [21], liver [22] and the spleen [23] with similar molecular architecture as seen in the central clock”. Why are examples limited exactly to these organs? At least, kidney (doi: 10.1681/ASN.2010080803), heart, fibroblasts, muscles, stomach, endocrine glands (https://doi.org/10.1016/j.neuron.2012.04.006; doi: 10.1146/annurev-neuro-060909-153128 ) brain and retina (doi: 10.1002/bies.20777) should not be omitted in this agenda.

Author response: We agree with reviews suggestion to include those important organs. We have now added them with the suggested paper in Page 3, line 129-130.

    P.2. L. 113-115: “CRDs are associated with abnormal CR of the lungs that reflect the variations in the daily changes of airway caliber, abnormal mucus secretion, increased airway resistance, insomnia, and decline in the lung function with abnormal immune-inflammatory responses [28].” I feel like mentioning insomnia logically does not belong here.

Author response: We agree with reviewer. Our sentence is particularly referring to various lung function parameter but not a separate set of disorder. So, we have now removed the word insomnia

    P.5. L. 175-176 – Please correct text formatting.

Author response: The unnecessary paragraph changes has been removed.

    P.9. L. 338 – Please explain the term OSA-18 as it is mentioned once, but never define

Author response: We have now elaborated the term OSA-18 as Obstructive sleep apnoea- 18 questionnaire in Page 9, Line 360

    P.9. L. 347 - Please explain the term TBPL-1 as it is mentioned once, but never defined.

Author response: We have explained the term TBPL-1 as TATA-box binding protein like 1

Location: page 9, line 369

    P.9. L. 347 - Please explain the term NRF-2 as it is mentioned once, but never defined.

Author response: We have mentioned the full name of NRF-2 as nuclear factor-erythroid 2-related factor, page 9, line 371

    P.10. L. 362 – “As CRs disruption promotes lung cancer progression, chrono-targeted chemotherapy treatment could be a promising therapeutic model for lung cancer treatment [75]”. Please define exactly, what kind of “CRs disruption” is meant here, since CRs disruption may have different causes: (artificial) light by night, shift-work, age-associated, clock gene mutations, etc.

Author response: According to the information provided in cited reference, we have now mentioned as As CRs disruption such as those cause by physiological perturbations (example jet lag), shift work, and mutation of central core clock gene promotes lung cancer progression, chrono-targeted chemotherapy treatment could be a promising therapeutic model for lung cancer treatment.

Location: page 9, line 385-386

    The term “personalized” chronotherapy was mentioned thrice with refs 76,96,98, though it never was substantiated. Authors are advised to define what they mean by “personalized chronotherapy” or refer to original concepts in the papers cited.

Author response: We have now elaborated the meaning of personalized therapy/chronotherapy as Until 2003, lung cancer patient was treated with palliative chemotherapy such as platinum based combination chemotherapy that improve survival by 8-10 months, while the recent introduction of personalized therapy allows to choose the best treatment option for specific lung cancer patient. This is possible by molecular testing such as next generation sequencing that provides information such as main oncogenic drivers (tumor promoter/suppressor), immune checkpoint that can be targeted with personalized cancer therapy. The administration of the personalized medications by aligning with persons circadian rhythm to improve treatment efficacy and tolerability (reduced side effects) is personalized chronotherapy.

Location: Page 10, line 399-407

    Overall:    While discussing optimal timing of delivery and timing of the expected target organ/tissue/cells response, please consider that it may vary extensively depending on broad in time individual differences in circadian phase, that may depend on the intrinsic clock or the individual adjustment to zeitgebers, the basics of personalized chronotherapy.

Author response: Indeed, the timing of delivery and cellular response to drugs depends on the individual difference in circadian phase and the oscillation of internal clock gene as well as individual adjustment to entrainments. We have mentioned a sentence after the explaining of personalized chronotherapy in the section Chronotherapy in lung cancer.

Location: page 10, line 407-411

    Also, while discussing “short-term”, and “long-term” drug formulas, please note that individual rates of absorption – distribution – metabolism – excretion (ADME) may vary dramatically between patients and depending on the time of delivery.

Author response: This is a very good suggestion and valid point to include. We have added few sentences on how the pharmacokinetics of drug vary between individual with an example between adults and pediatrics.

Location: Page 6, line 248-255

    I understand how tempting is the concept of standardizing timing for the given chemical – once and for all. However, personalized timing is highly important, since discarding adjustment of personalized phase differences in chronotherapy may negate its basic idea.

Author response: Personalized timing based on the biological diurnal rhythm of the patient show decreased side-effects and improved treatment success. Using machine learning and mathematical modelling approach, it is now possible to predict optimal time for delivery of chrono therapeutics. We have briefly discussed this in continuation of personalized therapy in Page 10.

    Since the paper’s title is “Recent advances in chronotherapy targeting respiratory diseases” paying at least some attention to chronotherapy of COVID-19 associated pulmonary disease, or drugs that may be repurposed to prevent COVID-19 caused pulmonary complications or combat its risks were expected to be found in this review.

Author response: We have now added a separate section 8. Chronotherapy in COVID-19

Location: Page 14-15, line 582-641.

Reviewer 2 Report

Authors described chronotherapy in respiratory diseases such as lung cancer and COPD. However, the contents are  not enough and needs improve the manuscript.

  1. In the first section, they described some clock genes CLOCK, BMAL1 etc. They need add bhlhe40/DEC1 and bhlhe41/DEC2 in the manuscript and Figure because these molecules are associated with pulmonary diseases. Also need discuss well.
  2. Circadian clock genes CLOCK, BMAL1 etc are well associated with cell cycle. They discuss well the correlation with circadian and cell cycle. If they would like to discuss chronotherapy, the correlation of clock genes and cell cycle is important. It has been reported that clock genes regulates cyclinD1 and c-Myc.
  3. Future perspectives should be improve. They should describe more realistic goal than unrealistic story.  

Author Response

Author response to reviewer comments

The authors are very thankful to the Editor, the Editorial team, and the Reviewers for consideration of our manuscript and for providing their valuable suggestions.

We have now addressed all the comments and incorporated the required changes into a revised version (highlighted in red font) as described in the point-by-point response below. These changes have helped to improve the manuscript substantially, which, we now hope is acceptable for publication in the Pharmaceutics journal.

Reviewer 2

Comments to the Author

Authors described chronotherapy in respiratory diseases such as lung cancer and COPD. However, the contents are not enough and needs improve the manuscript.

  1. In the first section, they described some clock genes CLOCK, BMAL1 etc. They need add bhlhe40/DEC1 and bhlhe41/DEC2 in the manuscript and Figure because these molecules are associated with pulmonary diseases. Also need discuss well.

Author response: We have now listed the bhlhe40/DEC1 and bhlhe41/DEC2 along with other gene in the introduction section. We have also revised figure 1 to include DEC1 and DEC2

Location of text: Page 2-3, line 102-113

Location of figure: page 3

  1. Circadian clock genes CLOCK, BMAL1 etc are well associated with cell cycle. They discuss well the correlation with circadian and cell cycle. If they would like to discuss chronotherapy, the correlation of clock genes and cell cycle is important. It has been reported that clock genes regulates cyclinD1 and c-Myc.

Author response: We agree with reviewer suggestion. As core clock gene influence the cell cycle progression by regulating the expression of cell cycle-related genes and thus participate in the development of tumors, we have discussed this in the section 5. Chronotherapy in lung cancer as

Location: Page 9, line 388-395

  1. Future perspectives should be improve. They should describe more realistic goal than unrealistic story.

Author response: We agree with reviewer suggestion. We have revised Future perspective section.

Page 15, line 644-661.

Round 2

Reviewer 2 Report

Authors well improved. I have no more claim.